# Cerebral Glucose Metabolism following TBI: Changes in Plasma Glucose, Glucose Transport and Alternative Pathways of Glycolysis—A Translational Narrative Review

**DOI:** 10.3390/ijms25052513

**Published:** 2024-02-21

**Authors:** Annerixt Gribnau, Mark L. van Zuylen, Jonathan P. Coles, Mark P. Plummer, Henning Hermanns, Jeroen Hermanides

**Affiliations:** 1Department of Anaesthesiology, Amsterdam UMC Location University of Amsterdam, Meibergdreef 9, 1105 AZ Amsterdam, The Netherlands; a.gribnau@amsterdamumc.nl (A.G.); m.l.zuylenvan@amsterdamumc.nl (M.L.v.Z.); j.hermanides@amsterdamumc.nl (J.H.); 2Department of Paediatric Intensive Care, Amsterdam UMC Location University of Amsterdam, Meibergdreef 9, 1105 AZ Amsterdam, The Netherlands; 3Division of Anaesthesia, Department of Medicine, University of Cambridge, Addenbrooke’s Hospital, Hills Road, Cambridge CB2 0QQ, UK; jpc44@cam.ac.uk; 4Intensive Care Unit, Royal Melbourne Hospital, 300 Grattan Street, Parkville, VIC 3050, Australia; mark.plummer@sa.gov.au

**Keywords:** traumatic brain injury, cerebral glucose metabolism, hyperglycaemia

## Abstract

Traumatic brain injury (TBI) is a major public health concern with significant consequences across various domains. Following the primary event, secondary injuries compound the outcome after TBI, with disrupted glucose metabolism emerging as a relevant factor. This narrative review summarises the existing literature on post-TBI alterations in glucose metabolism. After TBI, the brain undergoes dynamic changes in brain glucose transport, including alterations in glucose transporters and kinetics, and disruptions in the blood–brain barrier (BBB). In addition, cerebral glucose metabolism transitions from a phase of hyperglycolysis to hypometabolism, with upregulation of alternative pathways of glycolysis. Future research should further explore optimal, and possibly personalised, glycaemic control targets in TBI patients, with GLP-1 analogues as promising therapeutic candidates. Furthermore, a more fundamental understanding of alterations in the activation of various pathways, such as the polyol and lactate pathway, could hold the key to improving outcomes following TBI.

## 1. Introduction

Traumatic brain injury (TBI) is a growing public health problem with profound impact on individuals, their families, and society [1]. The majority of TBI-related emergency department visits, hospitalizations and deaths occur in young children aged 0–4 years, older adolescents aged 15–24 years, and older adults of 75 years and older [2]. More than one-third of patients with severe TBI do not survive to hospital discharge [3]. Severity is determined by the Glasgow Coma Scale (GCS) at presentation, with a GCS of 8 or lower classified as severe TBI and a GCS of 9 to 12 classified as moderate TBI [3]. Twenty-five percent of survivors live with severe disability across multiple domains including neurocognitive impairments, physical disabilities, psychiatric problems, and problems in social and work life [4,5].

Following the initial event, a cascade of molecular pathways is initiated in the subsequent hours and days, leading to secondary injury [6]. As the initial trauma cannot be reversed, medical care is directed at stabilization, resuscitation, and minimizing secondary injury [7]. Mechanisms involved in this secondary injury include damage from excitotoxicity, excess formation of reactive oxygen species (ROS), mitochondrial dysfunction, inflammation, apoptosis, axonal degradation, and metabolic derangements [6,8]. Disturbed glucose metabolism is a contributary factor in this secondary damage, with various studies linking disturbances in glucose metabolism to poor outcome [9].

In this review, we aim to provide an overview of the current literature on cerebral glucose metabolism following TBI. First, a concise overview of glucose metabolism in the normal functioning brain will be presented. Second, we will address glucose metabolism alterations in TBI patients, delving into alterations in plasma glucose, its transport into the brain, and the intricate pathways through which glucose is metabolised within the brain. Our objective is to elucidate the diverse array of changes that transpire following TBI and to identify potential targets for future therapeutic interventions within the context of this complex, heterogeneous, and debilitating condition.

## 2. Glucose Metabolism in the Healthy Brain

The brain is a metabolically demanding organ, constituting only 2 percent of total body weight, though utilizing 20 percent of the total available glucose supply at rest. Under physiological conditions, glucose is the primary energy source for the brain, and neurons use most of this energy [10,11]. Energy is utilised for various functions, including glutamate receptor activity, action and resting potentials, and transmitter release and recycling [12]. Maintaining these complex functions necessitates a continuous supply of glucose due to the brain’s limited storage capacity and limited alternative energy sources [8,13].

### 2.1. Transport

The blood–brain barrier (BBB) plays a crucial role in regulating cerebral substance influx and efflux. Glucose transport over the BBB is facilitated by transporters, primarily the glucose (GLUT) transporters (Figure 1). Among the 14 identified GLUT-transporters, GLUT-1 and GLUT-3 are the most dominant in cerebral glucose transport [14,15]. Both are insulin-, sodium-, and energy-independent transporters, which facilitate bidirectional transport [16,17].

GLUT-1 facilitates uptake across the BBB and into astrocytes [16,17]. Two different isoforms of GLUT-1 exist. They differ in glycosylation, but not in kinetics [17]. The 45 kDa isoform is present on astrocytes, and the 55 kDa isoform on the BBB [17]. Under resting conditions, the GLUT-1 transporter functions at less than its maximal capacity [11]. GLUT-3, mainly located on neurons, exhibits a high-affinity for glucose and ensures sufficient glucose uptake by neurons [15]. The other GLUT-transporters have less prominent roles in the brain. Other transporters, such as the sodium-dependent glucose transporters (SGLT), are also present on the BBB but are thought to have a minimal role under physiological conditions [18].

### 2.2. Different Routes of Glucose Metabolism

Once inside the cell, glucose is converted to glucose-6-phosphate by hexokinase, trapping it inside the cell. In aerobic conditions, glucose-6-phosphate undergoes subsequent conversion to pyruvate, which in turn undergoes oxidative decarboxylation to acetyl-CoA. Acetyl-CoA enters the citric acid cycle and through oxidative phosphorylation generates adenosine triphosphate (ATP). There are three other important pathways of glucose metabolism, which are summarised in Figure 1. First, the pentose-phosphate pathway, which is activated from glucose-6-phosphate primarily in response to the cellular demands for nucleic acid synthesis and the generation of NADPH, which is an important electron donor that ensures the preservation of antioxidants in their functional reduced state. Second, the production of glycogen from glucose-6-phosphate, which in the brain is solely produced in astrocytes [8]. Glycogen functions as an energy reserve, but concentrations are much lower in the brain compared to the liver or muscle, and can provide energy for only a few minutes [19,20]. Last, the conversion of pyruvate into lactate. Once thought to be toxic and solely produced under anaerobic conditions, the role of lactate in cerebral metabolism is thought to be more nuanced [8]. The astrocyte-neuron lactate shuttle (ANLS) is a widely discussed concept, where astrocytes take up glucose, convert it to lactate, and transport it to neurons by monocarboxylate transporters for use lactate in aerobic metabolism [8,9,21]. During activity, the neurotransmitter glutamate triggers increased uptake of glucose by astrocytes [21]. Studies indicate that neurons preferentially take up lactate over glucose as an oxidative substrate when both are available [21].

Alternative substrates for the brain are ketone bodies, which are byproducts of fatty acid metabolism. The brain converts ketone bodies to acetyl CoA for entry into the citric acid cycle, and such metabolism is predominantly observed during prolonged fasting states [22].

In the subsequent sections, we will explore the alterations in glucose metabolism following TBI, with a specific focus on changes in plasma glucose, variations in transport mechanisms, and the various pathways that glucose undergoes once inside the cell.

## 3. Alterations in Plasma Glucose following TBI

### 3.1. Plasma Glucose Levels following TBI

Hyperglycaemia is common in critically ill patients, even in those without a history of diabetes [23]. This is particularly pronounced in patients with severe TBI (GCS ≤ 8), with up to 87% of patients exhibiting hyperglycaemia upon admission [24]. There is also an association between the severity of TBI and plasma glucose levels: patients with more severe TBI tend to have higher plasma glucose values compared to patients with mild TBI [24,25,26,27]. This raises a critical question: are elevated glucose levels a mere indicator of TBI severity, or does it contribute to unfavourable outcomes?

Numerous studies have linked early hyperglycaemia to worse clinical outcomes [24,26,27,28,29,30,31,32]. These studies mostly investigated plasma glucose values in the first 24 h post-injury in relation to outcome [24,26,27,29]. In a retrospective study by Salim et al., it was found that persistent hyperglycaemia, defined as an average daily blood glucose ≥ 8.3 mmol/L in the first week, was present in 12.6% of patients with severe TBI (Abbreviated Injury Score (AIS) of ≥3) and was found to be an independent risk factor for death with an odds ratio of 4.91 (CI 2.88–8.56, *p* < 0.001) [30].

### 3.2. Mechanisms of Hyperglycaemia in TBI

#### 3.2.1. Stress Response

There are various mechanisms driving hyperglycaemia in TBI patients [9]. One of the main mechanisms is so-called stress-induced hyperglycaemia. Here, activation of the sympathetic nervous system and the hypothalamus-pituitary-adrenal (HPA) axis leads to increased release of catecholamines, growth hormone, cortisol and cytokines promoting hepatic glucose release by gluconeogenesis and other mechanisms [16]. A contributary factor is concomitant insulin resistance [16]. Impairment of post-receptor insulin-binding pathways and downregulation of GLUT-4, which facilitates peripheral (i.e., skeletal, cardiac, adipose) insulin-dependent glucose uptake and leads to decreased glucose utilization [16,33]. Cytokines play a role in this insulin resistance, with tumour necrosis factor-alfa (TNF-α) and interleukin-1 (IL-1) inhibiting post-receptor signalling, and epinephrine and cortisol reducing insulin-mediated uptake [16,34].

#### 3.2.2. Diabetes

Trauma patients with diabetes have a higher risk of mortality, and patients with insulin-dependent diabetes have a higher risk of mortality compared to non-insulin dependent diabetes [35,36]. Two studies distinguished stress-induced hyperglycaemia (SIH) from hyperglycaemia and diabetes, and found that patients with SIH have an increased risk of mortality compared to non-diabetic normoglycaemic patients. Diabetic hyperglycaemic patients, however, showed no increased risk of mortality [37,38]. In an intensive care unit (ICU) population, it was also found that prior exposure to hyperglycaemia reduced the association between glucose variability during the ICU stay and mortality compared to patients with no prior exposure to hyperglycaemia [39]. This suggests that glycaemic status before admission attenuates the impact of SIH.

#### 3.2.3. Inflammation

Following TBI, various cell types increase secretion of proinflammatory cytokines, activating multiple mechanisms leading to hyperglycaemia [34,40]. Specifically, cytokines such as IL-1, IL-6 and TNF-α are able to activate the HPA-axis, thereby elevating blood glucose levels [40]. Additionally, TNF-α contributes to insulin resistance, leading to hyperglycaemia [34].

#### 3.2.4. Iatrogenic Causes

Iatrogenic factors can induce hyperglycaemia. Surgical interventions are known to induce stress hyperglycaemia, with the extent depending on surgery type and severity [41,42]. Certain anaesthetics can also induce hyperglycaemia. General anaesthesia with intravenously administered anaesthetics exert a more profound impact on glucose metabolism compared to epidural anaesthesia with epidurally administered medication [42], and general anaesthesia with volatile anaesthetics having a more profound impact than total intravenous anaesthetics [43]. Furthermore, certain other types of medications, such as corticosteroids, can also induce hyperglycaemia [44]. The administration of substantial amounts of glucose through intravenous administration or enteral feeding can further contribute to hyperglycaemia [9].

### 3.3. Relationship between Plasma Glucose and Cerebral Glucose Levels

In the physiological state, cerebral glucose levels are approximately 20–80% of plasma glucose levels, and have a linear relationship with plasma glucose levels [8,45,46]. These values exhibit notable variations throughout the day, influenced by dietary intake [46].

In TBI patients, this linear relationship is mostly preserved [47,48,49]. Cerebral glucose is mainly measured by placing a microdialysis catheter in specific areas in the brain. Rostami et al. [49] placed two microdialysis catheters—one in the non-injured brain and one within an injured brain area—and revealed nuanced findings. In the non-injured brain, the positive relationship with plasma glucose measurements was sustained. However, in the injured area. 50% of the participants did not show any correlation between cerebral and plasma glucose values [49]. Another microdialysis study showed similar spatial variations, with a strong positive correlation between plasma glucose values and normal appearing brain and peri-penumbra. However, a weaker correlation was found between plasma glucose values and core and penumbra. [50]. It was argued that this might be explained by various metabolic changes in these areas [49].

### 3.4. Glucose Lowering Treatments

#### 3.4.1. Insulin

A pivotal single-centre study by van den Berghe et al. demonstrated reduced mortality with intensive insulin treatment targeting glucose levels of 4.4–6.1 mmol/L in critically ill patients [23]. However, subsequent multicentre trials in ICU patients, such as the NICE-SUGAR trial, revealed increased mortality with intensive glucose targets (4.1–6.0 mmol/L) compared to conventional glucose control (<10 mmol/L), and significantly more episodes of hypoglycaemia [51]. In a multicentre randomised-controlled trial comparing tight glucose control (4.4–6.1 mmol/L) and liberal glucose control (initiation of insulin treatment when glucose was >11.9 mmol/L) in ICU patients, a subgroup analysis revealed a potential link between lower mortality rates and tight glucose control specifically in patients with a neurologic or neurosurgical admission diagnosis [52]. They suggested that the brain-injured population might be particularly vulnerable to hyper- and hypoglycaemia [52]. A systematic review specific to TBI patients admitted to the ICU found no association between intensive glucose control and mortality, but noted a tendency towards improved neurological outcomes in intensive glycaemic control at the cost of increased hypoglycaemia [53]. Additionally, a microdialysis study revealed that tight glucose targets are associated with an increased rate of low cerebral glucose levels and a trend toward an increased incidence of critically low cerebral glucose levels [54].

#### 3.4.2. Glucagon-like Peptide-1 Analogues

Glucagon-like peptide-1 (GLP-1) is an incretin hormone normally produced in the intestine, where its release heightens in response to meals [55]. It enhances the sensitivity of the beta cells in the pancreas to glucose, leading to increased insulin release, thereby contributing to glucose homeostasis [55]. Additionally, it reduces hepatic glucose production by decreasing glucagon secretion in the alpha cells of the pancreas and slows gastric emptying [55]. Because of these mechanisms, GLP-1 analogues have proven effective in the treatment of type 2 diabetes. One of the advantages of this treatment is the low occurrence of hypoglycaemia [56].

GLP-1 receptors are also found in the central nervous system and are believed to play a role in regulation of appetite in the brain [56,57]. In vitro studies showed that GLP-1 analogues provided protection against oxidative stress, glutamate excitotoxicity, and reduced apoptosis—all significant factors contributing to the pathophysiology of TBI [56]. Animal studies showed upregulation of GLP-1 receptors post-TBI [58].

In rodent models of TBI, GLP-1 analogues have demonstrated neurotrophic and neuroprotective properties [56,59,60]. Furthermore, GLP-1 analogues exhibit anti-inflammatory actions by reducing glial cell activation and consequent cytokine production [59]. In a mouse model of mild TBI, a 7-day treatment with GLP-1 analogues resulted in reduced cognitive impairment up to 30 days after injury [56]. Also, improvement in sensorimotor function, oedema formation and BBB disruption are found after treatment with GLP-1 analogues [61]. With regard to glucose, it reduced glucose levels 30 min after TBI, without observing hypoglycaemia [61]. So far, no human studies have been conducted with GLP-1 analogues after TBI.

### 3.5. Future Research

In conclusion, glycaemic control after TBI is important, although the ideal plasma glucose target range is yet to be determined. This could also be a personalised target, depending on pre-admission glycaemic status, with GLP1 analogues as a possible adjunct to insulin treatment, whilst monitoring plasma as well as cerebral glucose levels if possible. Future research on glycaemic control after TBI should include these relevant factors.

## 4. Alterations in Glucose Transport following TBI

### 4.1. Glucose Transporters

#### 4.1.1. GLUT

The investigation of glucose transporter expression following traumatic brain injury (TBI) has predominantly been carried out in animal models. In animals, there is a consistent upregulation in GLUT-1 and GLUT-3 expression directly post-injury, indicative of heightened energy demands [62,63]. Nevertheless, different alterations in expression are reported in different glucose transporters. Most studies were conducted for GLUT-3 in neurons, and the 55 kDa isoform of GLUT-1, which is located at the BBB.

In rodents, increased expression of cerebral GLUT-3 was observed following injury, persisting from as early as 4 h after ictus until 48 h post-injury [64]. Zhou et al. [63] determined GLUT-3 concentration after TBI in rodents by measuring mRNA levels until 28 days after injury. After an initial surge 6 h post-injury, GLUT-3 mRNA concentrations in the injured area declined from day 3 onward, with low concentrations persisting in the cortex until day 7 and in the hippocampus until day 28. In non-injured areas, GLUT-3 mRNA concentration initially rises, returning to levels comparable to sham-operated rats at one day post-injury [63].

Interestingly, the 55 kDa GLUT-1 isoform had a different pattern [63]. At the injured area, mRNA levels of GLUT-1 also increased 6 h post-injury, but continued to display heightened expression levels up to 7 days post-injury in the cortex and up to 21 days in the hippocampus [63]. Within the non-injured area, GLUT-3 concentrations again showed an initial surge and returned to comparable levels of sham-operated rats on the first day post-injury [63]. The 55 kDa GLUT-1 expression was also investigated in two deceased patients at 7 and 8 h post-injury, through immunoreactivity by light and electron microscopy. Blood vessels close to the injured area showed a low expression of GLUT-1, in contrast to more distal areas where GLUT-1 was abundantly present [65].

In conclusion, there is an initial increase in expression of GLUT-1 and GLUT-3 post-injury. However, temporal and spatial variations may exist in expression patterns of GLUT-1 and GLUT-3 in the core and more distal areas.

#### 4.1.2. Sodium-Glucose Transporters (SGLT)

Sodium-glucose cotransporters are mainly known for their presence in the kidney, facilitating the transport of glucose along with sodium [66]. In healthy brain their expression levels and function is believed to be minimal [18]. Limited research has been done conducted in TBI models. In vitro and animal models of ischemia have demonstrated an increased expression and activity of SGLT [18]. In these stroke models, the inhibition of SGLT improved outcomes, reducing oedema associated with sodium transport [18]. In deceased patients with TBI, SGLT has been shown to be upregulated [67]. However, the precise role of these transporters in TBI and their potential effects remain to be elucidated.

### 4.2. Kinetics of Glucose Transport

As with expression of GLUT-transporters, spatial discrepancy was also observed in studies on kinetics in TBI patients [50,68]. Models of glucose transport are based on Michaelis-Menten kinetics, a mathematical concept stating that enzymatic reactions are saturable (Figure 2). According to this model, as substrate concentrations increase, the enzymatic reaction’s velocity rises until it reaches a maximum (Vmax), beyond which additional substrate does not further increase the reaction rate. The affinity of the transporter for the substrate is defined by the Michaelis–Menten constant (Km), representing the substrate concentration at which the reaction rate is half of Vmax.

Applying this to the brain, the model incorporates different compartments and relies on reversible Michaelis–Menten kinetics to facilitate transport between compartments (Figure 3). This allows a linear relationship between plasma and cerebral glucose [69]. In this model, k1 represents the rate constant of transport into the brain, while k3 represents hexokinase activity [68].

Two ^18^F-fluordeoxyglucose (FDG) positron emission tomography (PET) studies in TBI patients showed a reduction in k1 in and around the injured area [50,68]. In more distantly located areas, k1 was not reduced compared to healthy volunteers [50,68]. Concurrently, k3 was reduced on all sites, possibly because hexokinase is strongly inhibited by its product, assuming accumulation of glucose-6-phosphate [68]. This reduction in k3 may therefore be indicative of an overall reduction in activity [68].

### 4.3. BBB Disruption

The mechanical impact of TBI affects various structures in the brain, including vessels and different cell types, which contributes to impairment of the BBB. This disruption has three primary consequences [70]. First, it leads to increased BBB permeability due to damage to tight junctions, resulting in heightened transport of molecules that would otherwise be restricted from entering the brain [70]. Second, there is an elevation in transcytosis, a mechanism normally maintained at a low rate at the BBB. In TBI, this elevation contributes to the overall “leaking” of the BBB [70,71]. Last, cerebral oedema forms due to the development of vasogenic and cytotoxic oedema. Vasogenic oedema arises from an increased concentration of proteins and electrolytes in the exudate, leading to increased water extravasation [72]. Cytotoxic oedema, on the other hand, results from the failure of the Na/K pump in TBI, causing water accumulation within the cells and further disruption of the BBB [70,72].

In TBI, BBB disruption occurs in two phases. The early phase involves BBB disruption due to the mechanical forces during the initial trauma, while the second phase, occurring around three days after the initial trauma, is associated with inflammatory processes [73]. Animal models of diffuse axonal injury (DAI) showed a possible contributing factor of hyperglycaemia in BBB breakdown by induction of inflammation [74]. Nevertheless, there is ongoing debate regarding the precise mechanisms of secondary BBB breakdown [73].

### 4.4. Future Research

In summary, TBI induces a variety of changes in the transport of glucose into the brain. The spatial and temporal variations in the expression of glucose transporters and kinetics of glucose transport highlight the need for region- and time-specific analyses in future research. These analyses are also of interest when investigating the implications of glucose-lowering therapies. Additionally, examining the pathophysiology of BBB breakdown in a region-specific manner is crucial, with implications of examining hyperglycaemia, inflammation and other secondary insults as potentially modifiable factors.

## 5. Alterations in Cerebral Glucose Metabolism following TBI

### 5.1. Temporal Changes in Cerebral Glucose Metabolism

Changes in cerebral glucose metabolism following TBI typically manifest in two distinct phases: an initial hyperglycolytic phase followed by a period of hypometabolism.

The hyperglycolytic phase was first observed through an FDG-PET study by Bergsneider et al. revealing that post-injury there is an elevation of glucose metabolism without a concurrent rise in oxygen utilization [75]. Hyperglycolysis can occur in the absence of ischemia, and result from a variety of pathophysiological processes (inflammation, excitotoxicity, mitochondrial dysfunction) and shunting of glucose to other pathways [76]. This process, observed in approximately 56% of TBI patients in the first week post-injury [75], was confirmed by subsequent studies [76,77,78,79,80]. Multiple ^18^F-FDG-PET studies showed regional differences in hyperglycolysis, with hotspots mainly around the lesion site [50,68,75]. The duration of this phase is typically several days [81], and exists possibly because of the need to restore the ion gradients disrupted by the mechanical impact [80].

Subsequently, a phase of hypometabolism follows, persisting for weeks to months post-injury [76,80,81]. This metabolic depression mostly occurs without concurrent ischemia [76]. Spatial variations have been observed, with one study indicating reduced cerebral metabolic rate of glucose (CMRgluc) in grey matter compared to white matter in TBI patients, possibly due to reduced hexokinase activity [82]. Studies examining the relationship between consciousness and CMRgluc yield varying results. Bergsneider et al. reported no correlation between global cortical CMRgluc measured with ^18^F-FDG-PET and consciousness, measured by GCS-score. They highlighted a case where a conscious patient exhibited the same global cortical CMRgluc as a mechanically ventilated and sedated individual [78]. However, a subsequent study by Hattori et al. showed significant correlation between GCS-score before PET-scan and CMRgluc in the thalamus, brain stem and cerebellum [83].

As anaesthetics are also known to cause a reduction in cerebral glucose metabolism and cerebral blood flow [68,84,85], the reduction in CMRgluc observed in these studies likely results from a combination of pathophysiological and pharmacological effects [86]. In the following subsection, we will look at alterations in alternative pathways to glycolysis following TBI.

### 5.2. Changes in Alternative Pathways to Glycolysis

#### 5.2.1. The Pentose Phosphate Pathway (PPP)

The PPP is an oxygen- and ATP-independent metabolic pathway that yields various end products [87,88]. One of these products is ribose-5-phosphate, which is necessary for DNA repair and mRNA synthesis [77,87,88,89]. Additionally, the PPP produces NADPH, serving as an electron donor to preserve certain antioxidants in its reduced state [77,87,88,89]. Moreover, the PPP plays a role in fatty acid synthesis, neurotransmitter production, cholesterol metabolism, and can produce lactate [87].

Several studies used this formation of lactate to distinguish lactate produced by glycolysis or by the PPP by administering labelled 1,2-13C2 glucose. Animal studies showed an increase in lactate production from the PPP in TBI rats in comparison to control rats, while glycolysis remained dominant [77,89]. Studies in patients showed an increased flux into the PPP after TBI by measuring labelled lactate concentrations [87]. The PPP flux relative to glycolysis in TBI patients was 19.6% compared to 6.9% in healthy controls (*p* = 0.002) [87]. A microdialysis study revealed no significant difference in PPP-flux measured by labelled glucose between TBI patients and control patients undergoing surgery for benign tumours. Also in this study, glycolysis remained the dominant pathway and was elevated compared to controls. However, the trend suggested an elevated PPP-flux when cerebral oxygen concentrations decreased [88].

Collectively, these studies suggest increased glucose flux into the PPP after TBI. This may indicate increased demand for the products of the PPP or an existing pathological mechanism precluding normal glycolysis [87]. Notably, existing studies primarily focused on lactate as the product of the PPP and have not yet investigated other products of the PPP.

#### 5.2.2. Lactate

Historically, lactate is viewed as a waste product produced under anaerobic conditions [90]. However, over time, a shift in perception regarding the role of lactate in cerebral metabolism happened. Gallagher et al. demonstrated for the first time that lactate could be used as a substrate in the citric acid cycle in humans by delivering (13)C-labelled substrates in the brain and measuring (13)C-labelled metabolites using nuclear magnetic resonance [91].

In a cerebral microdialysis study of post-TBI patients, there were frequent episodes (15 to 30% of measurements) of extracellular glucose levels falling below a critical threshold [80]. The aetiology of these low glucose levels suggest an increased utilization of glucose or reduced supply. Notably, no correlation was found between episodes of low glucose and ischemic insults. A subsequent combined microdialysis and (FDG)-PET study confirmed a low incidence of ischemia post-TBI, which did not correspond with the levels of metabolic distress examined [92]. Interestingly, during episodes of low extracellular glucose, there was no observed increase in lactate concentration. Instead, it was seen that when glucose values dropped, lactate values also dropped. They called this the lactate paradox, and proposed that lactate was possibly being used as an alternative fuel after TBI [80,92], a possibility supported by other studies [20,81,90,93].

After TBI, there are also increased lactate levels outside of the brain [94]. Glenn et al. showed that after TBI, gluconeogenesis from this systemically elevated lactate is 67.1% in patients compared to 15.2% in healthy controls (*p* < 0.03). They also showed that the brain takes up this peripherally produced lactate, contributing to the hypothesis that lactate is used as an alternative fuel after TBI [24].

#### 5.2.3. Glycogen

Glycogen functions as an energy reserve, and is exclusively stored in astrocytes in the brain. Because of its limited storage capacity, it can be rapidly depleted within minutes [19,20]. However, a study employing a TBI model in rats, through lateral fluid percussion, revealed an increase in glycogen levels 24 h post-injury, specifically on the ipsilateral side. The authors hypothesised that this increase may represent a protective mechanism against ischemic events [95].

#### 5.2.4. The Polyol Pathway

An alternative metabolic pathway to glycolysis is the polyol pathway. Under normal circumstances, it constitutes a minor route due to the significant lower affinity of aldose reductase (AR) (the initial enzyme) for glucose compared to hexokinase [96]. However, under specific circumstances, such as hyperglycaemia, this pathway can be activated [96,97]. In the polyol pathway, glucose is converted to sorbitol and fructose with potential cyto- and neurotoxic properties (Figure 4). It can induce axonal damage through osmotic activity and diminish resilience against oxidative stress [96,97]. It is thought to be a contributor to complications of diabetes mellitus, such as neuropathy [96].

In the brain, activation of the polyol pathway is associated with white matter damage and cognitive impairments in psychiatric patients and children with diabetes mellitus type 1 [96,98]. In animal models of ischemic events, AR knockout mice showed lesser neurological deficits and smaller infarct sizes in comparison with controls. Depletion of sorbitol dehydrogenase (SDH) did not change neurological deficit or infarct size [99]. Also in a rodent model of spinal cord contusion, inhibitors of AR and SDH resulted in an improvement in locomotor function [100]. To our knowledge, no studies have investigated the activation of the polyol pathway in TBI patients thus far. Considering these findings, one could hypothesize that the polyol pathway may also contribute to secondary injury in TBI.

### 5.3. Future Research

In conclusion, TBI induces dynamic shifts in cerebral glucose metabolism, characterised by an initial hyperglycolytic phase followed by a prolonged hypometabolic phase, accompanied by spatial variations. Upregulation of alternative pathways, such as the pentose phosphate pathway and lactate production, is observed. Further research is warranted to precisely understand the roles of both pathways after TBI. Additionally, the polyol pathway may contribute to secondary injury and needs investigation in TBI patients.

## 6. Clinical Aspects of Cerebral Glucose Metabolism in TBI

Approaches such as microdialysis and magnetic resonance spectroscopy provide insight into cerebral glucose metabolism. However, translating results from animal models to clinical settings is challenging, given the diversity observed in TBI [101]. Ongoing efforts in metabolic disruption research are focused on improving therapeutic approaches and improving the overall prognosis for people with TBI.

Microdialysis is facilitating clinical research on TBI and its relationship to cerebral glucose metabolism. Studies integrating ^18^F-FDG-PET and cerebral microdialysis show disturbed glucose metabolism in TBI [102]. However, as discussed in this review, there is a wide range of factors affecting cerebral glucose metabolism in TBI, underscoring the importance of examining trends rather than individual values. Clinically, one must realise that only small areas of the brain are sampled when using microdialysis, resulting in a lower sensitivity of the findings.

Carbon-13 (^13^C) labelling studies offer an alternative way to investigate the clinical facets of TBI and their correlation with glucose metabolism. Unlike conventional radioisotope labelling, the stable isotope ^13^C functions as an efficient tracer for glucose [102]. Methods such as nuclear magnetic resonance (NMR) spectroscopy and gas chromatography-mass spectrometry (GC-MS) examine downstream metabolites and provide valuable insights into metabolic pathways [102]. Studies administering ^13^C-labeled glucose to rat models with TBI show altered cerebral glucose metabolism, implying a potentially substantial restorative and protective role for the damaged brain [87]. Combining ^13^C-labeled substrate microdialysis with ex vivo NMR may provide a reliable and cost-effective way to assess cerebral glucose metabolism [102]. Understanding the changes in cerebral glucose metabolism in TBI may help develop targeted therapeutic strategies to promote brain repair in TBI patients.

## 7. Conclusions

Glucose metabolism is markedly deranged following traumatic brain injury, and likely contributes to secondary brain injury and adverse outcomes. Mechanisms driving dysregulation are dynamic, and contribute to early hypermetabolism and later hypometabolism. The optimum plasma glucose target remains controversial, with some evidence favouring tighter control, making the brain-injured population distinct from the broader critically ill population.

## Figures and Tables

**Figure 1 ijms-25-02513-f001:**
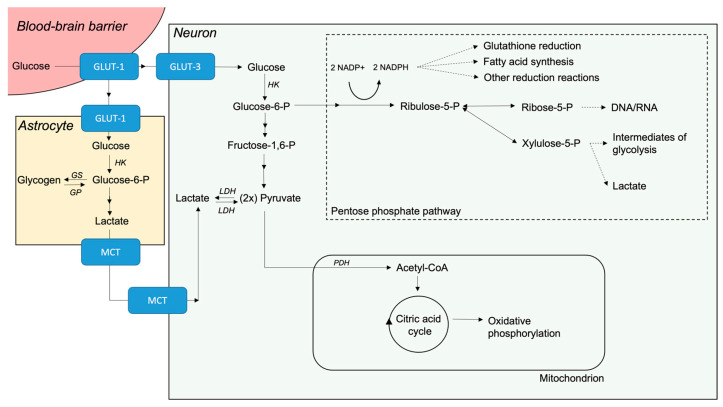
Schematic summary of cerebral glucose metabolism in the healthy brain. Abbreviations: GLUT, glucose transporter; GP, glycogen phosphorylase; GS, glycogen synthase; HK, hexokinase; LDH, lactate dehydrogenase; MCT, monocarboxylate transporter; PDH, pyruvate dehydrogenase.

**Figure 2 ijms-25-02513-f002:**
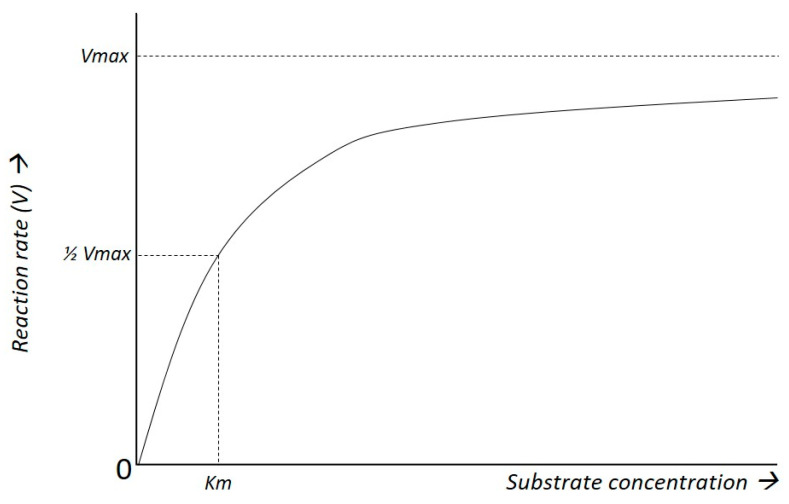
Michaelis–Menten kinetics. As substrate concentration rises the reaction rate increases until it approaches the maximum reaction rate (Vmax). The Michaelis–Menten constant (Km) is defined as the substrate concentration at which the reaction rate is half of Vmax.

**Figure 3 ijms-25-02513-f003:**
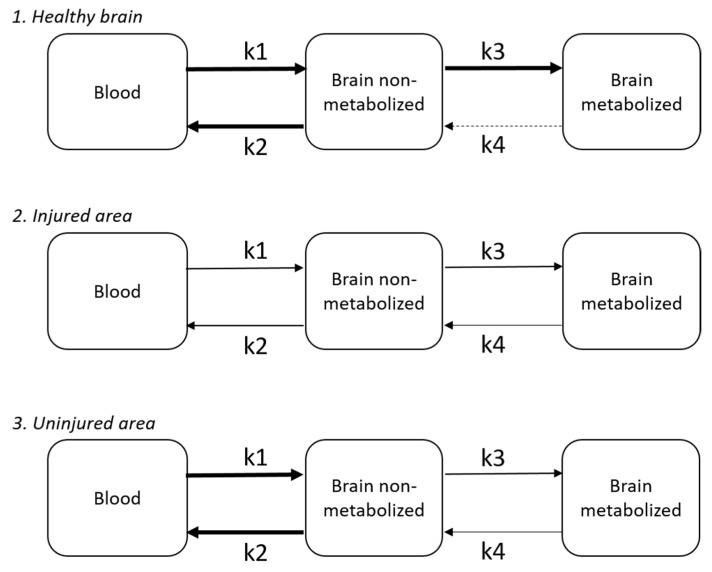
Schematic representation of reversible Michaelis–Menten kinetics in brain compartments. After TBI, a decrease is seen in the injured area (2) in k1 and k3 compared to the healthy brain (1). In the uninjured area after TBI (3), only a decrease in k3 is seen. Figure based on measurements by Hattori et al. [68]. k1 represents transport of glucose into the brain by glucose transporters, whereas k3 represents hexokinase activity.

**Figure 4 ijms-25-02513-f004:**
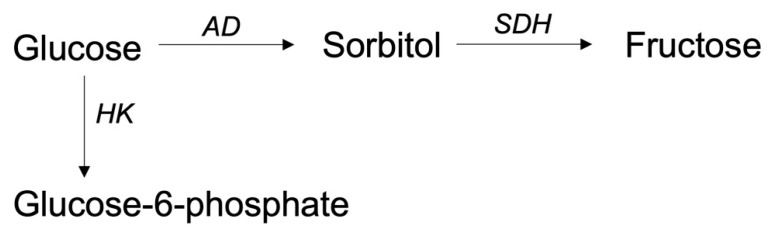
Polyol pathway. Abbreviations: AD, aldose reductase; HK, hexokinase; SDH, sorbitol dehydrogenase.

## Data Availability

No new data were created or analysed in this study. Data sharing is not applicable to this article.

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
