# Peer review of "Cerebral Glucose Metabolism following TBI: Changes in Plasma Glucose, Glucose Transport and Alternative Pathways of Glycolysis—A Translational Narrative Review"

_ijms, 2024, doi:10.3390/ijms25052513_

Round 1
Reviewer 1 Report
Comments and Suggestions for Authors
The authors provided a systematic review of cerebral glucose metabolism post TBI. They described the mechanism of hyperglycemia post TBI in detail and included the metabolism characteristics post TBI in this paper. This review is impressive and would be helpful for research in this field. I have two concerns:
Line 62-65: You mention that the energy in the cortex was predominantly consumed by glutamate receptors (50%), AP (21%), RP (20%), and transmitter release/recycling (9%). Although you cite a reference to support your opinion, the energy consumption in the brain is far more complicated, and it is extremely difficult to trace the specific energy consumption in the CNS. Thus, I propose not to mention the accurate percentage number here.
Figure 1: Although it summarizes glucose metabolism in the brain well, the font size in this figure is too small. Could you make some modifications?"
Reviewer 2 Report
Comments and Suggestions for Authors
In this manuscript, Gribnau et. al. have compiled an impressive review of the role of glucose metabolism in TBI. However, there are several areas that can be improved:
- The authors can improve their title and emphasize the clinical aspects underlying the translational relevance of their analyses.
- The authors can improve Figure 2 with TBI-specific inputs.
- The authors need to improve Figure 3 and add specific physiological and perturbed states (also capitalized, k or K?).
- The authors need to expand after citing literature to provide perspective on these points throughout the manuscript.
- The authors need to strengthen the clinical component of the TBI and glocose metabolism, such as : Jalloh I, Carpenter KL, Helmy A, Carpenter TA, Menon DK, Hutchinson PJ. Glucose metabolism following human traumatic brain injury: methods of assessment and pathophysiological findings. Metabolic brain disease. 2015 Jun;30:615-32.
The language needs minor revision for flow and cohesiveness.
Round 2
Reviewer 2 Report
Comments and Suggestions for Authors
The authors have improved the manuscript.